# Cross-Video Pedestrian Tracking Algorithm with a Coordinate Constraint

**DOI:** 10.3390/s24030779

**Published:** 2024-01-25

**Authors:** Cheng Huang, Weihong Li, Guang Yang, Jiachen Yan, Baoding Zhou, Yujun Li

**Affiliations:** 1School of Geography, South China Normal University, Guangzhou 510631, China; huangch@m.scnu.edu.cn (C.H.); liweihong@m.scnu.edu.cn (W.L.); 2021023461@m.scnu.edu.cn (J.Y.); 2023023398@m.scnu.edu.cn (Y.L.); 2Guangdong Shida Weizhi Information Technology Co., Ltd., Qingyuan 511500, China; 3SCNU Qingyuan Institute of Science and Technology Innovation, Qingyuan 511500, China; 4College of Civil and Transportation Engineering, Shenzhen University, Shenzhen 518060, China; bdzhou@szu.edu.cn

**Keywords:** cross-video, coordinate features, pedestrian tracking, overlapping view

## Abstract

Pedestrian tracking in surveillance videos is crucial and challenging for precise personnel management. Due to the limited coverage of a single video, the integration of multiple surveillance videos is necessary in practical applications. In the realm of pedestrian management using multiple surveillance videos, continuous pedestrian tracking is quite important. However, prevailing cross-video pedestrian matching methods mainly rely on the appearance features of pedestrians, resulting in low matching accuracy and poor tracking robustness. To address these shortcomings, this paper presents a cross-video pedestrian tracking algorithm, which introduces spatial information. The proposed algorithm introduces the coordinate features of pedestrians in different videos and a linear weighting strategy focusing on the overlapping view of the tracking process. The experimental results show that, compared to traditional methods, the method in this paper improves the success rate of target pedestrian matching and enhances the robustness of continuous pedestrian tracking. This study provides a viable reference for pedestrian tracking and crowd management in video applications.

## 1. Introduction

Surveillance videos play a vital role in public safety and personnel management by monitoring and capturing real-time data within their coverage areas. With improvements in information construction in various scenes, the expanse of surveillance video coverage has significantly increased. Pedestrian tracking in videos is an important task for various industries, but it remains challenging due to numerous obstacles. Video tracking [1] mainly includes three parts [2,3,4]: pedestrian detection [5,6,7], pedestrian matching [8,9], and pedestrian trajectory acquisition [4,10].

In current research, although there are many studies on pedestrian tracking, these methods are mostly limited to extracting pedestrian information solely from a video context. Establishing a connection between monitoring videos and the real environment requires personnel familiar with both, but accurately describing the location of the target pedestrian remains challenging. Meanwhile, in many application scenarios, such as crime investigation and security management, obtaining real-time location information of pedestrians is crucial for precise tracking and monitoring.

Because of the limited coverage of single surveillance cameras [11], current research focuses on employing multiple cameras for continuous target tracking [12]. Cross-video pedestrian tracking refers to tracking pedestrians in multiple different surveillance videos, achieving consistent correlation across multiple video frames to enable the seamless tracking of pedestrians across different surveillance devices. In cross-video pedestrian tracking, a common approach involves extracting pedestrian appearance features using neural networks for similarity assessment. However, relying solely on pedestrian appearance features can be challenging for the accurate identification of target pedestrians due to factors such as environmental lighting, camera angles, and clarity.

Additionally, when a target pedestrian performs actions across multiple videos in overlapping fields of view, often only one surveillance camera’s pedestrian location is chosen while disregarding the other, failing to fully utilize the information resources of multiple surveillance devices. Furthermore, due to inherent limitations in tracking algorithms, surveillance devices, and their placement, there may be errors in pedestrian localization. The experiments in Refs. [13,14] have also indicated that as pedestrians move away from the center of the camera’s field of view, localization errors tend to increase, resulting in pedestrian position jumps and trajectory discontinuities [15,16,17].

Consequently, in order to address these aforementioned challenges, this paper proposes a cross-video pedestrian tracking algorithm with a coordinate constraint that effectively utilizes spatial information to complete a pedestrian tracking cross-video. Specifically, instead of traditional image-based tracking trajectories, this paper proposes a collaborative real coordinates approach to reflect the trajectory information of pedestrians. Moreover, this paper introduces spatial affinity to balance the process of pedestrian matching, which only relies on appearance similarity. Because of the poor robustness and low accuracy of cross-video tracking, a linear weighted method is proposed. This method utilizes multiple location data for data association, thereby achieving a more robust transition across videos.

The contributions of this paper are as follows:The introduction of spatial affinity improves the accuracy and reliability of pedestrian re-identification.The proposed linear weighted algorithm enhances the precision and robustness of cross-video tracking effectively.

The remainder of this paper is structured as follows: Section 2 provides related studies. Section 3 outlines the principles of related methods and the construction of the proposed cross-video pedestrian tracking algorithm with coordinate constraints. Subsequently, Section 4 presents our experimental scheme and experimental results, followed by a comprehensive analysis and discussion of the experimental results in Section 5. Our conclusion is given in Section 6.

## 2. Related Work

### 2.1. Pedestrian Localization

Based on the revelation of a paper by [18], a convolutional neural network is used to extract feature maps in the spatial domain. Accurate pedestrian spatial localization information in video streams and images is a fundamental basis for the comprehensive analysis of pedestrian behavior [19]. Meanwhile, real-world pedestrian positions are crucial factors in management and decision making. Lie Yang et al. [20] employed a fisheye camera mounted on an indoor ceiling for detecting and tracking pedestrian heads, ensuring consistency in pedestrian trajectories between the video and the real world. Additionally, depth cameras have been utilized to obtain three-dimensional information [21], providing another avenue for video-based pedestrian positioning. Zoltan Koppanyi et al. [22] achieved real position estimation without camera calibration through triangulation and DEM-related principles. Domonkos Varga et al. [23] inferred the actual positions of pedestrians based on image registration and matching. However, the exploration of tracking real-world pedestrian position information in video content remains limited despite various estimation methods for pedestrian location positioning.

### 2.2. Across-Video Pedestrian Tracking

When cross-video pedestrian tracking is performed, pedestrian matching steps are required first. A common approach is to utilize the invariance of pedestrian feature information. An improved multi-person matching cascade scheme was proposed to solve these problems by Yundong Guo et al. [24]. It can increase the accuracy of inter-camera person re-identification (Re-ID) by taking advantage of association priorities. Qiang Wang et al. [25] introduced a prototype-guided instance matching framework to boost the performance of multiple pedestrian tracking tasks, where we can jointly learn object detection and appearance embedding in an end-to-end fashion. However, in real-world scenarios, appearance variations in video pedestrians under different cameras or angles can disrupt the precise matching of pedestrian information, leading to tracking errors [26,27].

Cross-video pedestrian tracking is applied in two application scenarios as follows: overlapping views and nonoverlapping views [28]. In the case of nonoverlapping views, prevailing strategies rely on the similarity of pedestrian features across different surveillance sources [29]. By contrast, investigations into pedestrian tracking within an overlapping view have explored various solutions. For instance, Liu Caihong et al. [30] achieved large-scale, long-distance pedestrian tracking by seamlessly merging overlapping regions from multiple surveillance videos. Sohaib Khan and Mubarak Shah [31] introduced the concept of the camera’s field of view (FOV) boundary as a foundational premise for cross-video tracking. The overlapping view boundaries are defined as the boundaries of areas where two or more visual fields overlap. Building upon this foundation, Yang Xia et al. [32] leveraged the scale-invariant feature transform (SIFT) algorithm to autonomously identify matching points and generate FOV lines when pedestrians traverse the overlapping view boundary. This approach facilitates smooth transitions between two cameras for pedestrian tracking. Furthermore, Xinchao Xu et al. [33] employed an overhead camera angle to capture pedestrians, ensuring that the pedestrian’s size remained consistent during movement in the video. This characteristic simplifies continuous tracking across cameras by capitalizing on the scale invariance of pedestrian size for matching. However, the universal applicability of overhead cameras is constrained, and the available pedestrian feature information is restricted, posing challenges for pedestrian matching. Most cross-video pedestrian tracking studies focus primarily on automatically generating and dividing FOV boundaries to resolve location conflicts. However, these studies tend to overlook the potential of multidimensional position information from overlapping views to improve the robustness and accuracy of pedestrian tracking.

## 3. Materials and Methods

The overall framework is presented in Figure 1. Firstly, this study processes videos from two surveillance cameras with overlapping views. In Camera A, target pedestrian X is detected, while Camera B identifies potential candidates. Subsequently, the matching process for the candidates in Camera B is conducted by combining appearance features and coordinate features, accurately locating target pedestrian X. Upon the matching of pedestrian X using both Camera A and Camera B, the linear weighted processing of pedestrian X’s position is executed using auxiliary lines within the overlapping view. Ultimately, the system outputs the complete trajectory of target pedestrian X.

### 3.1. Mapping the Coordinates of Pedestrians in the Video

#### 3.1.1. Coordinate System Transformation

In order to better describe the trajectory of pedestrians and establish a connection between the video and the actual world, we perform coordinate system transformations. The principle of coordinate system transformation derives from the field of photogrammetry [34,35]. By deducing the relationship between the world and the pixel coordinate system, we can achieve the functionality of obtaining the real-world position of pedestrians from a two-dimensional video frame. The processes for converting pixels into world coordinates are presented as follows:The internal and external parameters of the camera are obtained using camera calibration, such as the focal length of the camera, pixel size, rotation matrix, and the translation vector of the camera. In this paper, Zhang Zhengyou’s checkerboard camera calibration method [36] is used to obtain the internal and external parameters of the surveillance.The relationship between the pixel and the image coordinate system is a process of continuation [37].The relationship between the camera and the image coordinate system is based on perspective and can be calculated using the principle of similar triangles.The relationship between the world and the camera coordinate system is a rigid transformation involving rotation and translation.

Combining these steps, we can obtain the equation for converting similar coordinates to world coordinates as follows:(1)$$z\left[\begin{array}{c} u\\ v\\ 1 \end{array}\right]=\left[\begin{array}{ccc} f/dx & 0 & u_{0}\\ 0 & f/dy & v_{0}\\ 0 & 0 & 1 \end{array}\right]\left[\begin{array}{cc} R_{3*3} & t_{3*1}\\ O_{1*3} & 1 \end{array}\right]\left[\begin{array}{c} X_{w}\\ Y_{w}\\ Z_{w}\\ 1 \end{array}\right]$$
where *u*_0_ and *v*_0_ represent the coordinates of the pixel coordinate system’s origin in the horizontal and vertical directions; *dx* and *dy* represent the lengths of the pixel grid in the horizontal direction and vertical direction, respectively; *f* represents the focal length of the camera; (*X_w_*, *Y_w_*, *Z_w_*) represents the world coordinates corresponding to the pixel coordinates of the pedestrian (*u*, *v*); *R*_3∗3_ represents a 3∗3 rotation matrix; and *t*_3∗1_ represents a 3∗1 translation vector.

#### 3.1.2. Coordinate System Unity

After projection transformation, two surveillance systems each establish their own three-dimensional coordinate systems. Based on these corresponding points, unity between the two coordinate systems is achieved using rotation and translation matrices.

We selected four corresponding points *P_Ai_* (*x_Ai_*, *y_Ai_*, *z_Ai_*) and *P_Bi_* (*x_Bi_*, *y_Bi_*, *z_Bi_*) from the overlapping view of Cam_A and Cam_B (where *i* = 1, 2, 3, 4). These coordinates are represented in matrix form as shown in Equations (2) and (3):(2)$$A=\left[\begin{array}{cc} \begin{array}{cc} x_{A1} & y_{A1}\\ x_{A2} & y_{A2} \end{array} & \begin{array}{cc} z_{A1} & 1\\ z_{A2} & 1 \end{array}\\ \begin{array}{cc} x_{A3} & y_{A3}\\ x_{A4} & y_{A4} \end{array} & \begin{array}{cc} z_{A3} & 1\\ z_{A4} & 1 \end{array} \end{array}\right]$$
(3)$$B=\left[\begin{array}{cc} \begin{array}{cc} x_{B1} & y_{B1}\\ x_{B2} & y_{B2} \end{array} & \begin{array}{cc} z_{B1} & 1\\ z_{B2} & 1 \end{array}\\ \begin{array}{cc} x_{B3} & y_{B3}\\ x_{B4} & y_{B4} \end{array} & \begin{array}{cc} z_{B3} & 1\\ z_{B4} & 1 \end{array} \end{array}\right]$$

As shown in Equations (4) and (5), the transpose matrix of *B* is multiplied by matrix *A* to obtain matrix *M*. Then, singular value decomposition [38] is performed on *M*, resulting in the left singular vector *U*, the right singular vector *V*, and the singular value matrix *S* [39]. Using the first three columns of *U* and *V*, we construct *R* (rotation matrix). By subtracting the average coordinates of *B* from the average coordinates of *A*, we obtain the translation vector t. This computation yields the final transformation matrix *T*, which achieves the unity of the two coordinate systems.
(4)$$M=B^{T}A=USV^{T}$$
(5)$$\left\{\begin{array}{c} R=UV^{T}\\ t=mean\left(B\right)-mean\left(A\right) \end{array}\right.\Rightarrow T=\left[\begin{array}{cc} R & T\\ 0 & 1 \end{array}\right]$$

### 3.2. Cross-Video Pedestrian Tracking Method

#### 3.2.1. Traditional Pedestrian Detection and Matching

YOLOv3 [5,6,7] was released by Joseph Redmon from the University of Washington in April 2018. Compared to other object detection algorithms, YOLOv3 requires fewer computational resources and has lower hardware requirements, rendering it widely applicable in the field of surveillance [40,41]. YOLOv3 adopts Darknet-53 as its backbone network, and the loss function is defined by the bounding box loss, confidence loss, and classification loss. Since the objects we mainly focus on are pedestrians, the classification loss can be disregarded. Therefore, the loss function equation is presented as follows:

The bounding box loss is expressed by Equations (6) and (7) as follows:(6)$$L_{xy}=\lambda_{coord}\sum_{i=0}^{S^{2}}\sum_{j=0}^{B}1_{ijk}^{obj}{\left(x_{i}^{pred}-x_{i}^{ture}\right)}^{2}$$
(7)$$L_{wh}=\lambda_{coord}\sum_{i=0}^{S^{2}}\sum_{j=0}^{B}1_{ijk}^{obj}{\left(w_{i}^{pred}-w_{i}^{ture}\right)}^{2}$$

The confidence loss is expressed by Equation (8) as follows:(8)$$L_{conf}=\sum_{i=0}^{S^{2}}\sum_{j=0}^{B}1_{ijk}^{obj}{\left(IOU-{\hat{p}}_{i}^{obj}\right)}^{2}+\lambda_{noobj}\sum_{i=0}^{S^{2}}\sum_{j=0}^{B}1_{ijk}^{noobj}{\left(IOU-{\hat{p}}_{i}^{obj}\right)}^{2}$$
where *L_xy_* represents the loss for the predicted bounding box center coordinates, and *L_ωh_* represents the loss for the predicted bounding box’s width and height. $x_{i}^{pred}$ and $w_{i}^{pred}$ represent the predicted box’s center coordinate and width–height, respectively, while $x_{i}^{true}$ and $w_{i}^{true}$ represent the true box’s center coordinate and width–height, respectively. *S* represents the size of the feature map, *B* represents the number of predicted bounding boxes per position, and *k* represents the position index on the feature map. *λ_coord_* is a hyperparameter used to adjust the weight of the bounding box’s coordinate loss. *L_conf_* represents the confidence loss, *IOU_i_* represents the intersection over the union between the predicted box and the true box and ${\hat{p}}_{i}^{obj}$ is the probability of the predicted box containing a pedestrian. *λ_noobj_* is a hyperparameter that is used to adjust the weight of the confidence loss for non-pedestrian regions.

The basic components of a convolutional neural network (CNN) [42] include convolutional layers, pooling layers, and fully connected layers [43]. The convolutional layer aims at extracting local information using convolutional operations, starting from individual pixels and gradually expanding the receptive field [44]. The pooling layer is responsible for down-sampling or up-sampling the input data to reduce the computational complexity. The fully connected layer maps the output feature maps to specified classification results via matrix calculations [45]

The principal equation for the convolutional layer is given by Equation (9):(9)$$p_{y\left(i,j\right)}=\sum_{m}\sum_{n}p_{x}\left(i-m,j-n\right)*h\left(m,n\right)$$

The principal equation for the pooling layer is given by Equation (10):(10)$$\left\{\begin{array}{c} y\left(i,y\right)=pooling\left[\theta \right]\\ \theta =x_{\left(i\times s\right):\left(i\times s+R-1\right),\left(j\times s\right):\left(i\times s+R-1\right)} \end{array}\right.$$

The principal equation for the fully connected layer is given by Equation (11):(11)$$y=W^{T}x+b$$

In the equations, *p_x_*(*i*,*j*) represents the input pixel, *h*(*m*,*n*) represents the weight of the convolution kernel, *p_y_*(*i*,*j*) represents the output pixel, *s* represents the stride, and *R* represents the pooling window size. *W* represents the weight matrix, *b* represents the bias vector, *x* represents the input feature vector, and *y* represents the output vector.

The triplet loss function [46] is utilized to build a better model. By minimizing the distance between the same pedestrians and maximizing the distance between different pedestrians, the loss function equation is established, as shown in Equation (12):(12)$$Loss=k*{\left(1-\cos\left(A,B\right)\right)}^{2}+\left(1-k\right)*\max{\left[1-\left(1-\cos\left(A,B\right)\right),0\right]}^{2}$$
where *k* is the label, which can have a value of either 0 or 1; (1 − cos(*A*,*B*)) represents the distance score between the two feature vectors; and *m* is the constant used to control the threshold of similarity. If *k* = 1, we aim to minimize the Euclidean distance between the two feature vectors. Conversely, if *k* = 0, we aim to maximize the Euclidean distance between the two feature vectors.

#### 3.2.2. Pedestrian Matching by Introducing the Coordinate Constraint

Pedestrian matching is another challenging continuous tracking task, but the prevailing matching methods only rely on the similarity among the pedestrian’s appearance features, leading to incorrect results. The position of the pedestrian in the video is objectively determined; theoretically, the coordinates of the same pedestrian in both cameras A and B are identical at the same moment. Therefore, this paper proposes using the coordinate features of pedestrians to constrain the matching process. As described in “Section 3.1.1 Coordinate System Transformation”, the position points of all pedestrians in video B are obtained through photogrammetric principles. The affinity between the target pedestrian and the candidate pedestrian is calculated based on the Euclidean distance difference between their positions. Tracking errors may occur in the system due to the influence of the camera’s clarity, color, and frame rate [47,48]. Based on the test results, the error outcomes roughly follow a normal distribution. Since the appearance similarity and coordinate error distribution probability are independent of each other, the product of them represents the matching probability. The equation is expressed as shown in Equation (13):(13)$$\left\{\begin{array}{c} \varepsilon \left(i\right)=\sqrt{{\left(x_{t}-x_{i}\right)}^{2}+\left(y_{t}-y_{i}\right)}\\ P=max\left[f\left(\varepsilon \left(i\right)\right)P_{cnn}\left(i\right)\right] \end{array}\right.$$

Between the target pedestrian and candidate pedestrian *i*, *ε*(*i*) represents the Euclidean distance, *f*(*ε*(*i*)) is the probability density function value of the normal distribution corresponding to the Euclidean distance, *P_cnn_*(*i*) is the appearance similarity extracted using the CNN, and *P* is the probability of the candidate pedestrian who matches the target pedestrian. The equation states that *f*(*ε*(*i*)) is multiplied by *P_cnn_* to achieve a balance. Specifically, when there are many pedestrians in the video, and the reliability of appearance similarity decreases, using this equation improves the accuracy and reliability of pedestrian matching.

#### 3.2.3. Cross-Video Pedestrian Tracking with Linear Weighting

The channel and spatial reliability tracker (CSRT) algorithm [49] is a mature, widely applicable, and high-performance object-tracking algorithm [50]. Thus, the CSRT algorithm has good real-time performance and practical application. The core formula is given by Equation (14):(14)$$\delta =\left[dx,dy,ds,dp\right]$$

Here, *dx* and *dy* represent the changes in the pedestrian’s center, *ds* represents the change in the pedestrian’s size, and *dp* represents the change in the pedestrian’s rotation angle. This algorithm uses a regression model to predict the pedestrian’s state in the next frame based on a training dataset, enabling pedestrian tracking.

However, despite its real-time performance and practicality, the CSRT algorithm is associated with low accuracy. To decrease the tracking error, inspired by the part of “*Image Fusion Module*” in the literature [51], this paper proposes a linear weighting method for weighted auxiliary lines, as shown in Figure 2 (where Center_A and Center_B represent the principal points of photographs in surveillance A and surveillance B, respectively; A and B represent the points at the connection of Center_A and Center_B, respectively, that is, the baseline, intersects on the boundary lines of the overlapping view; *Pedestrian*_*X_A_* and *Pedestrian*_*X_B_* represent the tracking positions of pedestrians in surveillances A and B; and *X_A_* and *X_B_* represent the orthogonal projections of *Pedestrian*_*X_A_* and *Pedestrian*_*X_B_*, respectively, on the baseline). By considering the two positions of the target pedestrian obtained from the two surveillances, the position closer to the principal point of the photograph is selected for the linear weighting step. This method assesses the confidence level when localizing target pedestrian X in surveillances A and B, which relies on the proportion of pedestrian X who occupies the length of line AB on View_AB. As pedestrian X moves beyond the field of view of surveillance A, the level of confidence in localizing pedestrian X in surveillance A gradually decreases to zero. This trend ensures the robustness of the entire pedestrian transition process.

The principal equation for linear weighting of the weighted auxiliary line is given in Equation (15):(15)$$\left\{\begin{array}{c} P_{X}=k*P_{XA}+\left(1-k\right)*P_{XA}\\ k=\left|BX\right|/\left|AB\right| \end{array}\right.$$
where the selection process of X is completed in X_a_ and X_b_, and its selection criteria are based on the minimum distance between the two and their corresponding monitoring center point; |*BX*| and |*AB*| represent the lengths of the line segments *BX* and *AB*, respectively; *k* represents the linear weighting factor; *P_XA* and *P_XB* represent the positional information obtained from the tracking of pedestrian *X* through surveillance A and B, respectively, at the same moment; and *P*_*X* represents the position of the final pedestrian *X*.

#### 3.2.4. Cross-Video Pedestrian Tracking Algorithm Workflow

In this study, the principles of photography and the CSRT algorithm are used to realize the continuous tracking of pedestrians on the Cam_A surveillance screen and to calculate the coordinates of pedestrians. When pedestrian X enters the overlapping view of two surveillances, the candidate pedestrian in Cam_B is detected and matched via a coordinate feature and similarity. In the overlapping view, the weight k is adjusted according to the position of pedestrian X on the weighted auxiliary line. When pedestrian X is closer to surveillance A, k is increased to place more trust in the position information of surveillance A. Conversely, when pedestrian X is closer to surveillance B, k is reduced to place more trust in the location information of surveillance B, and pedestrian tracking cross-video is realized.

The specific algorithm steps are described as follows:Pedestrian X is detected and tracked using the CSRT algorithm to acquire its position in each frame.The coordinates of the two surveillance systems are unified using an overlapping view, principles of photography, and rotation and translation matrices.When pedestrian X enters the overlapping view, surveillance A obtains the position of X, and the candidate pedestrians in surveillance B are matched using the positional features and the features extracted via the neural network.Using the position of pedestrian X that projects onto the weighted auxiliary line, weights are assigned to determine the positions of pedestrian X obtained from surveillances A and B.The weights of both surveillances are adjusted as the distance of pedestrian X changes from Cam_A to Cam_B, enhancing the tracking robustness of pedestrians between the two surveillances.

The algorithm workflow is illustrated in Figure 3.

## 4. Results

### 4.1. Experimental Protocol

#### 4.1.1. Experimental Scene

To validate the proposed cross-video pedestrian tracking algorithm, an experiment was conducted in the circular square in front of the Guangzhou Science and Technology Achievement Transformation Center, as shown in Figure 4. Here, labeled points *P_Ai_* and *P_Bi_* (*i* = 1, 2, 3, 4) in the figure serve as homonym-labeled reference points for unifying the coordinate system, and *P_A5_*, *P_A6_*, *P_B5_*, *P_B6_* are points to validate that the two surveillances achieve coordinate unification. The radius of the square is approximately 15 m (m). There are seven pedestrians in the square as experimental subjects.

The weighted auxiliary line is established by setting identical markers and establishing a unified coordinate system. Since the position of a pedestrian is determined based on their foot placement while moving, the experimental results are presented in terms of planar information without considering *Z*-axis coordinate data. According to Equations (1), (4), and (5), the transformed coordinates of the principal point of the photograph are (159.1, 368.4) in surveillance A and (1703.5, 361.3) in surveillance B (unit: centimeters). The weighted auxiliary line intersects with the boundaries of the surveillances at (481.9, 366.9) and (1116.3, 364.0). The principal points of the photograph of the two surveillances and the weighted auxiliary line are shown in Figure 5.

Since the Y-direction error between endpoint A and endpoint B of the weighted auxiliary line is within 10 cm, for the sake of the calculation’s convenience, it is approximated as a line segment with a constant Y value.

#### 4.1.2. Experimental Hardware and Parameters

The two surveillances employed in this experiment are *Hikvision Ezviz H5* surveillances, as shown in Figure 6. The focal length is 4 mm, and the resolution is 2 million pixels. Surveillances A and B are positioned 15 m from the center of the square, with a height of approximately 4 m. The Euler angles for surveillance A are a pitch angle of 10 degrees, a yaw angle of 15 degrees, and a roll angle of 0 degrees. For surveillance B, the Euler angles are a pitch angle of 10 degrees, a yaw angle of −15 degrees, and a roll angle of 0 degrees. After calibrating the cameras using Zhang Zhengyou’s camera calibration method [36], the transformation parameter of surveillance is approximately $\left[\begin{array}{ccc} 674.90 & 0.0 & 479.06\\ 0.0 & 672.08 & 280.63\\ 0.0 & 0.0 & 1.0 \end{array}\right]$, where the focal length in the x-direction *fx* = 674.90, the focal length in the y-direction *fy* = 672.08, and the coordinates of the pixel center point are (479, 280). The rotation matrix for surveillance A is $R\_A=\left[\begin{array}{ccc} 0.9956 & 0.0621 & -0.0698\\ -0.0840 & 0.9222 & -0.3774\\ 0.0409 & 0.3816 & 0.9234 \end{array}\right]$, and the rotation matrix for surveillance B is $R\_B=\left[\begin{array}{ccc} 0.9659 & -0.2546 & 0.0405\\ -0.2846 & 0.9396 & -0.1420\\ 0.0794 & 0.1937 & 0.9776 \end{array}\right]$. The translation matrix for surveillance A is $R\_A={\left[\begin{array}{ccc} 7.6233 & 6.6627 & -2.3028 \end{array}\right]}^{T}$, and the translation matrix for surveillance B is $R\_A={\left[\begin{array}{ccc} 5.1222 & -10.9856 & -4.2732 \end{array}\right]}^{T}$.

#### 4.1.3. Pedestrian Tracking Performance Metrics

This paper primarily employs three evaluation metrics for the pedestrian tracking trajectory assessment, namely, the root mean square error (*RMSE*), mean error (*ME*), and maximum error (*MaxE*), as shown in Equations (16)–(18), respectively. *RMSE* is used to measure the average level of long-term error between predicted or estimated values and actual observed values, reflecting the overall stability of the tracking algorithm. *ME* provides the average deviation between predicted or estimated values and the actual observed values. In contrast to *RMSE*, it does not consider the polarity of the error but focuses solely on its magnitude. *MaxE* is used to reveal the maximum deviation between predicted or estimated values and the actual observed values. In trajectory evaluation, it can be utilized to assess the performance of the tracking algorithm in worse-case scenarios.
(16)$$RMSE=\sqrt{\frac{\Sigma_{i=1}^{n}{\left({\hat{x}}_{i}-x_{i}\right)}^{2}+{\left({\hat{y}}_{i}-y_{i}\right)}^{2}}{n}}$$
(17)$$ME=\frac{\sum_{i=1}^{n}\sqrt{{\left({\hat{x}}_{i}-x_{i}\right)}^{2}+{\left({\hat{y}}_{i}-y_{i}\right)}^{2}}}{n}$$
(18)$$MaxE=argmax\left[\sqrt{{\left({\hat{x}}_{i}-x_{i}\right)}^{2}+{\left({\hat{y}}_{i}-y_{i}\right)}^{2}}\right]$$

In the equations, (${\hat{x}}_{i},{\hat{y}}_{i}$) represents the true position of the pedestrian, while ($x_{i}$, $y_{i}$) represents the estimated position of the pedestrian.

### 4.2. Pedestrian Detection and Matching

For pedestrian detection, this study uses the YOLOv3 algorithm combined with the COCO class index to detect pedestrians. By filtering out other classes and retaining only the pedestrian class, pedestrian detection is accomplished. As shown in Figure 7, the blue boxes represent the candidate pedestrians, while the red boxes represent the target pedestrians. By introducing the coordinate features of pedestrians, the success rate of pedestrian matching can be improved.

Produced by Equation (13), the candidate pedestrian with the highest probability (*P*) is considered the target pedestrian. Using pedestrian 1 as an example, the matching probability is shown in Table 1, where the coordinates of the target pedestrian calculated in Monitor A are (456, 348). The mean value of the normal distribution is 20 cm, and the standard deviation is 15 cm, as obtained from the test data.

Here, *P_cnn_* represents the similarity of pedestrians, and *f*(*i*) represents the probability density function value of the error between candidate pedestrians and target pedestrians. The success rate when matching the 7 tested pedestrians using the traditional method is 5/7. The success rate after introducing coordinate constraints is 7/7.

To validate the higher accuracy of our method, it is important to note that our approach requires specific coordinate data to constrain the pedestrian-matching process. Publicly available datasets do not provide the coordinates of pedestrians. In this study, we utilized a custom dataset consisting of 20 pedestrians and 200 images captured from different angles and surveillances. The results of our validation are presented below in Table 2:

Through the comparison results, the pedestrian matching method proposed in this paper has better performance in mAP evaluation indicators than CNN.

### 4.3. Cross-Video Pedestrian Tracking Experiment Results

#### 4.3.1. Cross-Video Pedestrian Tracking without Coordinate Constraints

When performing cross-video tracking, pedestrian 1, who was tracked (highlighted with a red box in Figure 8A), is matched with pedestrian 3 (highlighted in a blue box in Figure 8B) with the highest similarity at the same moment. However, pedestrian 3 is located at a distance from pedestrian 1 in terms of coordinates, clearly indicating that they are not the same person. The resulting trajectories are shown in Figure 9.

From the trajectory plot above, while tracking pedestrian 1 from surveillance A to B within the overlapping view, there is an incorrect association with pedestrian 3, resulting in tracking errors and the loss of information regarding pedestrian 1.

#### 4.3.2. Cross-Video Pedestrian Tracking with Coordinate Constraint

Three pedestrians in the surveillance video are selected to display the results of this experiment. Among them, pedestrian 1 is matched in the overlapping view fields of the two surveillance videos, as shown in Figure 10. The trajectories of these three pedestrians are shown in Figure 11. The red trajectory lines represent the tracking trajectories of surveillance A to pedestrians; the blue trajectory lines represent the tracking trajectories of surveillance B to pedestrians; the black trajectory lines represent the real trajectories of pedestrians; and the green trajectory lines represent the pedestrian trajectories weighted by the weighted auxiliary line.

The weight is assigned according to the position of the pedestrian relative to the weighted auxiliary line. The complete trajectory after fusion is shown in Figure 12, where the red trajectory line is the complete trajectory of the merged pedestrian, and the black trajectory line is the real trajectory of the pedestrian.

From the above results, the pedestrian trajectories without weighted auxiliary lines (that is, traditional pedestrian tracking) exhibit varying degrees of position point jumps and trajectory line mutations. In contrast, the pedestrian trajectories obtained with weighted auxiliary lines closely align with the real trajectories without position point jumps or trajectory line mutations.

A comprehensive error analysis is conducted from the following four perspectives: single surveillance A, single surveillance B, average weighting, and auxiliary line weighting. This analysis focuses on the perpendicular distance between the position points obtained for each frame of the three pedestrians and the straight line representing the true trajectory, as shown in Figure 13. The line charts clearly demonstrate that using the linear weighting of the weighted auxiliary line method significantly improves positioning and tracking accuracy in terms of the mean error value, maximum error value, and root mean square error value. Our method is better than the average weighting method. Specifically, for pedestrian 1, the average error accuracy improves by 6.96 cm (cm), the maximum error accuracy improves by 8.95 cm, and the root mean square error accuracy improves by 3.27 cm. In the case of pedestrian 2, the average error accuracy improves by 12.72 cm, the maximum error accuracy improves by 43.33 cm, and the root mean square error accuracy improves by 9.06 cm. Similarly, for pedestrian 3, the average error accuracy improves by 6.75 cm, the maximum error accuracy improves by 11.85 cm, and the root mean square error accuracy improves by 2.79 cm.

## 5. Discussion

In this paper, we introduce an algorithm for cross-video pedestrian tracking with a coordinate constraint, which is designed to address the practical application needs of pedestrian tracking in multi-surveillance video scenarios. Experiments were conducted to demonstrate the feasibility and effectiveness of this method. In this paper, we discuss the reasons behind the experimental results obtained with this algorithm.

The clarity and color of the surveillance affect the appearance features of the monitored pedestrians, but by introducing coordinates as pedestrian feature information, the matching process not only follows similarity but also considers the closeness of the pedestrian locations. In addition, when monitoring numerous pedestrians, the probability of incorrect matching is greatly decreased.In monitoring the location of a pedestrian from multiple monitors, pedestrian tracking is not limited to one monitor and has strong robustness. Simultaneously, the weights of pedestrian locations are assigned based on a weighted auxiliary line, which weakens the values with large tracking errors and yields more precise results for pedestrian tracking.

Terminal-based positioning methods (ultrawideband [54], Bluetooth [55], wireless fidelity [56], etc.) are limited in practical scenes, such as places where people are not allowed to carry terminal equipment or label loss. Positioning using surveillance videos is expected to solve the above application problems. Cross-video pedestrian tracking is the premise of the practical application of positioning using surveillance videos. This paper proposes a cross-video pedestrian tracking method that not only accurately matches pedestrian targets (by introducing coordinate features) but also achieves the return of a continuous trajectory (by linear weighting). This paper provides viable technical support and a case study for large-scale, accurate, and continuous pedestrian tracking in practical application scenarios. However, this study still faces certain limitations. First, in the case of nonoverlapping viewing fields, it is hard to apply coordinate features to constrain pedestrian matching. Additionally, when targets are heavily occluded, there is currently no suitable solution to address this issue.

## 6. Conclusions

Due to the constraints of terminal-based and single-surveillance tracking methods, there has been a growing interest in adopting cooperative tracking methods in multi-surveillance systems. However, the traditional pedestrian reidentification algorithm faces challenges in cross-video matching. Additionally, camera performance and tracking algorithm limitations significantly hinder cross-video pedestrian tracking. To solve these problems, this paper presents a cross-video pedestrian tracking algorithm with a coordinate constraint. Our approach involves obtaining the real location of the target pedestrian via coordinate system transformation and pedestrian detection. Then, the pedestrian location serves as feature information to improve the success rate of the matching process. Throughout the cross-video pedestrian tracking process, weights are assigned to the obtained locations using weighted auxiliary lines, outputting more reliable locations. Experiments show that this algorithm achieves higher positioning accuracy and better robustness in cross-video pedestrian tracking. Accurate and continuous pedestrian tracking enhances the accuracy of pedestrian detection and behavior analysis in video surveillance scenarios, thereby improving the intelligence of monitoring systems. In practical applications, this study also strengthens criminal investigations and improves traffic management efficiency. However, this study only focuses on the pedestrian tracking process from an overlapping view. Future research should further investigate pedestrian tracking from a nonoverlapping view to resolve the persistent issue of continuous tracking under all conditions.

## Figures and Tables

**Figure 1 sensors-24-00779-f001:**
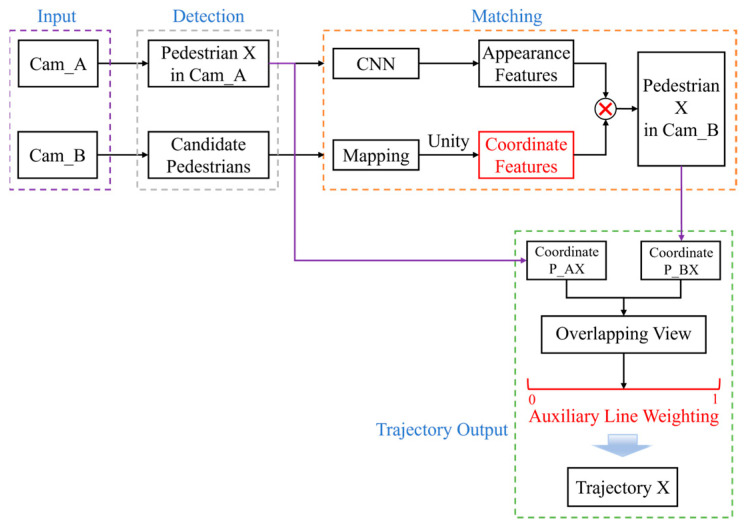
Framework of the proposed algorithm.

**Figure 2 sensors-24-00779-f002:**
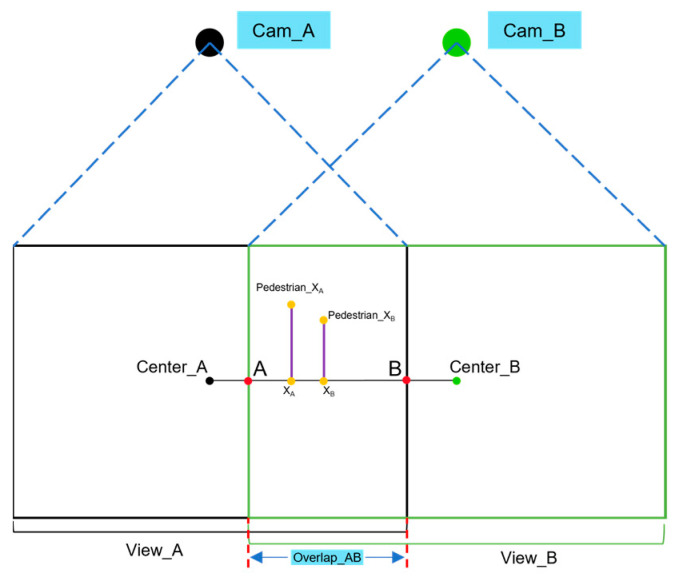
Illustration of weighted auxiliary line for pedestrian trajectory acquisition.

**Figure 3 sensors-24-00779-f003:**
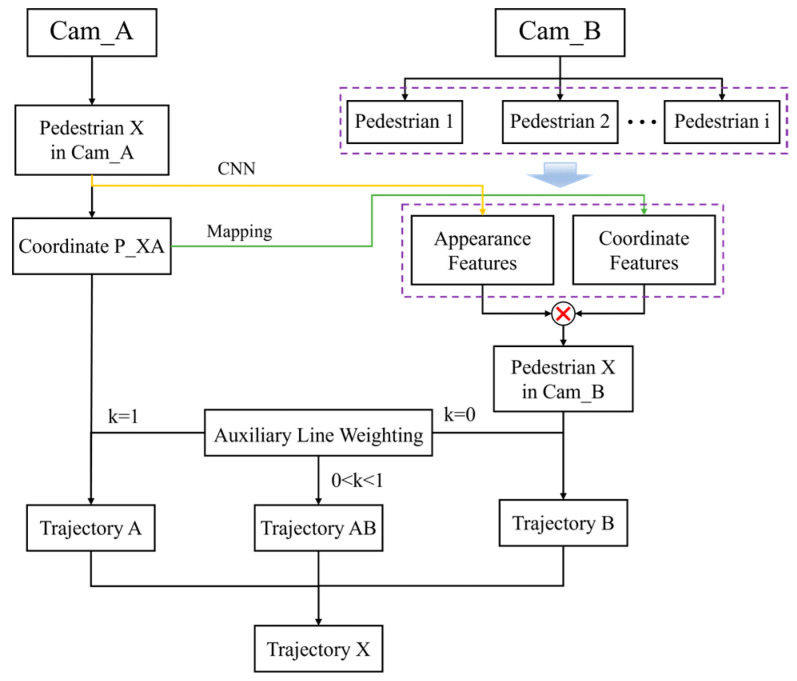
Flowchart of cross-video pedestrian tracking algorithm.

**Figure 4 sensors-24-00779-f004:**
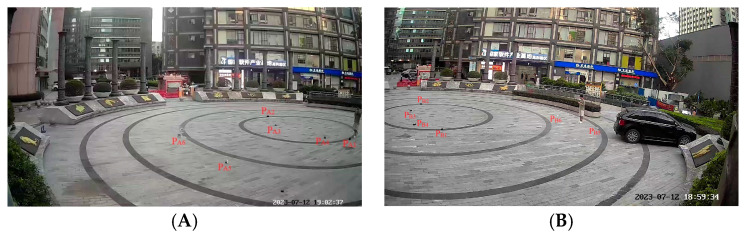
Experimental scenes. (**A**) Perspective view of surveillance, (**B**) Perspective view of surveillance.

**Figure 5 sensors-24-00779-f005:**
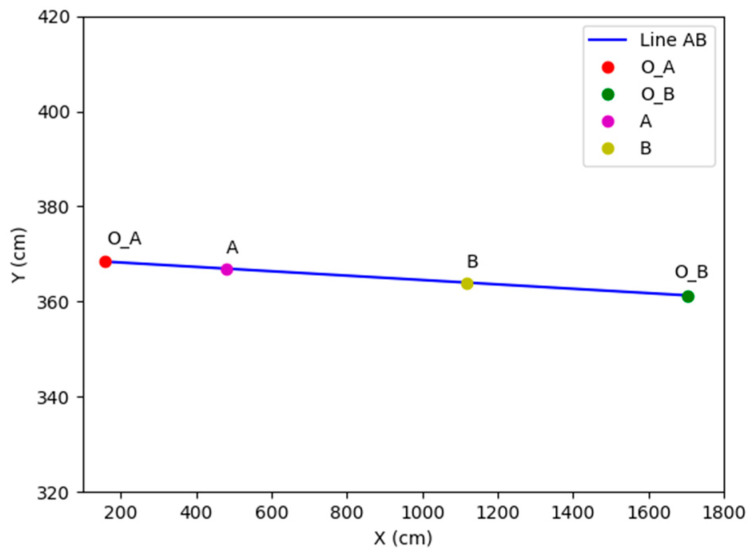
Weighted auxiliary line.

**Figure 6 sensors-24-00779-f006:**
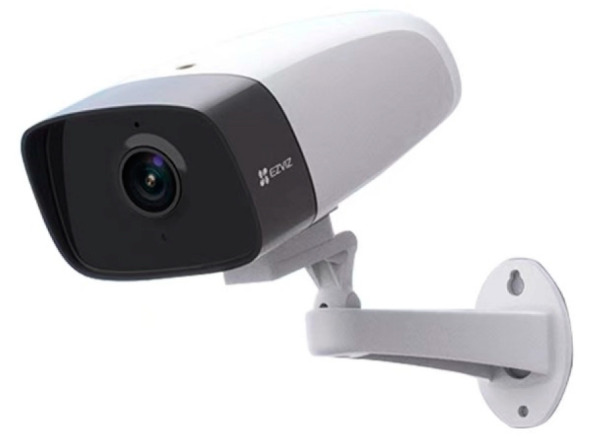
Hikvision Ezviz H5 surveillance.

**Figure 7 sensors-24-00779-f007:**
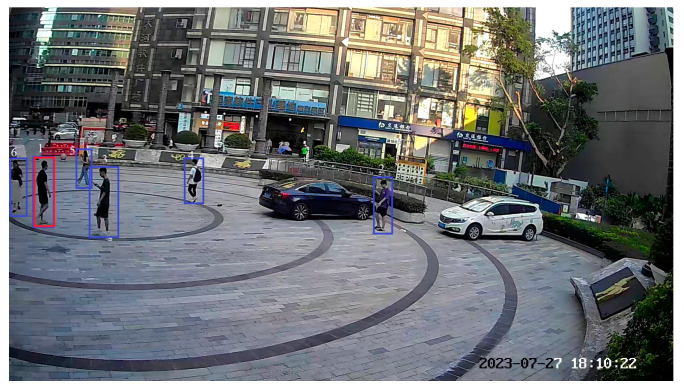
Pedestrian detection. Blue boxes: candidate pedestrians; red boxes: target pedestrians.

**Figure 8 sensors-24-00779-f008:**
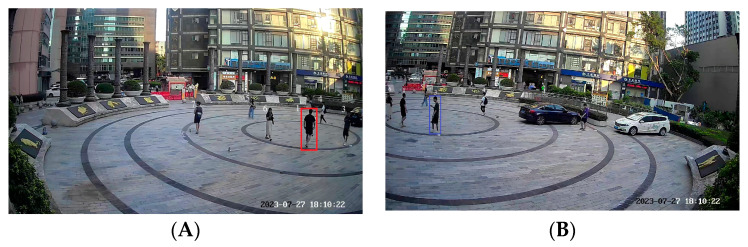
Cross-video pedestrian matching without coordinate constraints in the overlapping view. (**A**) Pedestrian 1 in surveillance, (**B**) Matching result in surveillance.

**Figure 9 sensors-24-00779-f009:**
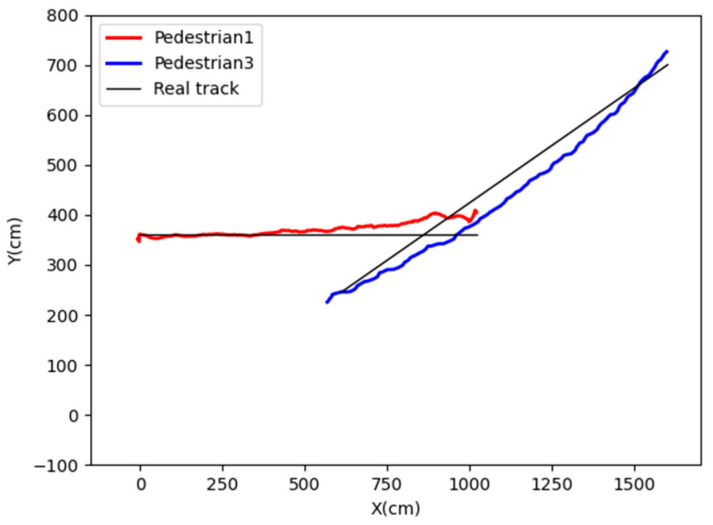
Trajectories of pedestrian 1 and pedestrian 3.

**Figure 10 sensors-24-00779-f010:**
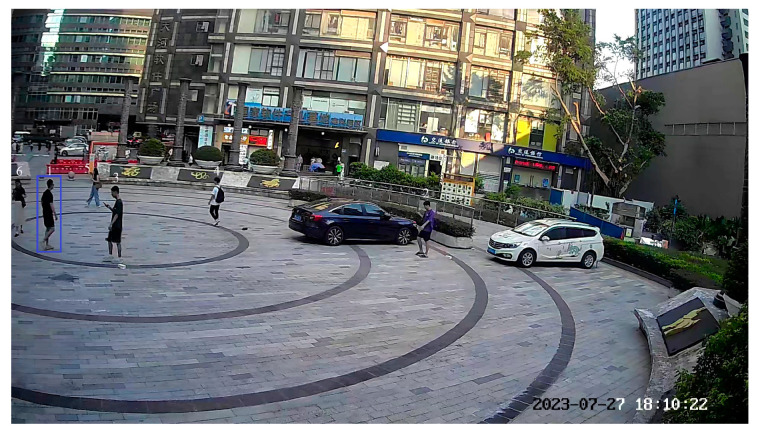
Matching the result of pedestrian 1 with coordinate constraints in the overlapping view.

**Figure 11 sensors-24-00779-f011:**
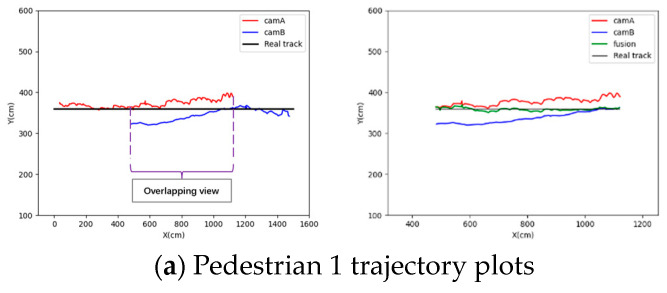
Cross-video pedestrian tracking with overlapping view (**Left**: complete trajectories of cam_A and cam_B; **Right**: details of the overlapping region and fusion trajectories).

**Figure 12 sensors-24-00779-f012:**
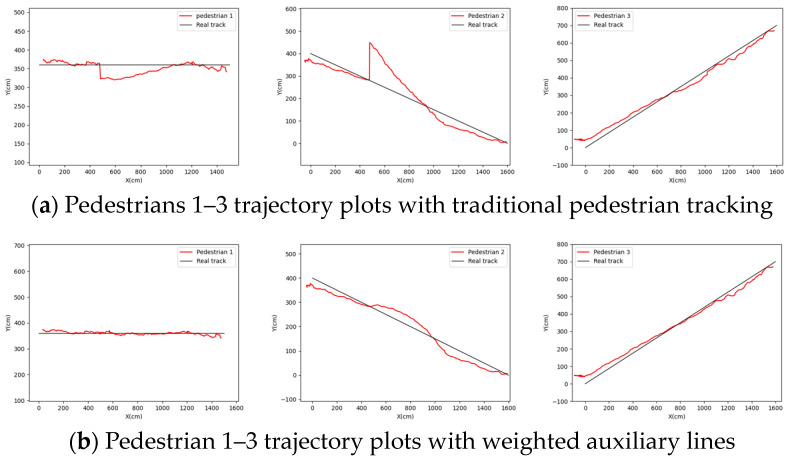
Complete trajectories of pedestrian tracking across videos.

**Figure 13 sensors-24-00779-f013:**
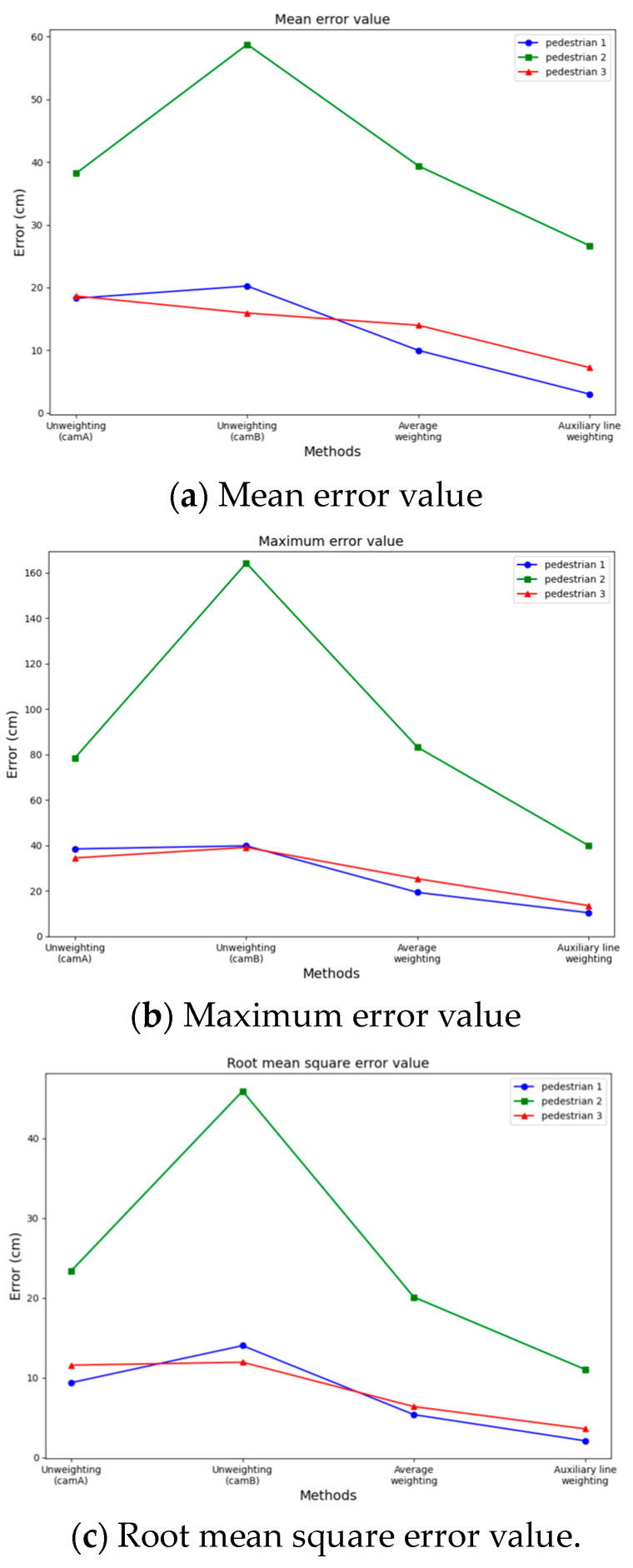
Pedestrian tracking error graphs.

**Table 1 sensors-24-00779-t001:** Pedestrians matching probability.

Candidate Pedestrians	Coordinates	*P_cnn_*	*f*(*i*)	*p*
1	(405, 345)	0.4140	**2.032 × 10^−2^**	**0.0084**
2	(1520, 284)	0.1570	0	0
3	(536, 308)	**0.4280**	1.281 × 10^−2^	0.0055
4	(986, 878)	7.249 × 10^−5^	4.033 × 10^−17^	0
5	(591, 1146)	0.0005	6.725 × 10^−22^	3.36 × 10^−27^
6	(386, 517)	7.199 × 10^−6^	9.216 × 10^−9^	6.64 × 10^−16^

**Table 2 sensors-24-00779-t002:** The mean average precision (mAP) of different algorithms.

Methods	mAP (%)
PCB [8]	64.8
GLAD [9]	61.3
AutoLoss-GMS-A [52]	70.5
PDC [53]	66.1
CNN [42]	52.6
Ours	**76.8**

## Data Availability

Data are contained within the article.

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
