# Peer review of "Cross-Video Pedestrian Tracking Algorithm with a Coordinate Constraint"

_sensors, 2024, doi:10.3390/s24030779_

Round 1
Reviewer 1 Report
Comments and Suggestions for Authors
The authors propose a pedestrian tracking method over multiple cameras in this manuscript. The topic of the manuscript is a hot research topic in the literature and fits well to the scope of Sensors journal.
I have the following comments and concerns on this manuscript.
i) Although pedestrian tracking over one or multiple cameras is a very hot research topic, the authors opted to not write a separate related works section. The introduction section is rather weak. The reader does not get information on the main methods of pedestrian tracking. I think the authors should split the first section into Introduction and Related works section. The review of the state-of-the-art is not complete. There are also methods which use uncalibrated cameras for tracking, i.e. A multi-view pedestrian tracking method in an uncalibrated camera network, ICCV, 2015 or Deriving Pedestrian Positions from Uncalibrated Videos, 2017. The authors should mention this line of works as well.
ii) Unfortunately, the contributions are very general. The authors write: "The trajectory of pedestrians is described using actual coordinates, effectively associating video content with the real world." What does actual coordinate mean in this context? What does "effectively" mean in this context?
iii) Unfortunately, Figure 1 is very small and probably does not contain all details of the proposed method. In general, all figure captions are very short and not informative. I think figure captions should be informative and self-contained.
iv) In general, the authors described well what was done. For me, the "*" symbol is very unusual in matrix formulas. I would use "X" (\times in Latex) instead. I think the authors described the weighting scheme carefully, although the formulas are difficult to understand probably due to poor formatting of equations. I think it would be better to write the manuscript in Latex instead of Word in case of writing such complex formulas.
v) I think the presentation of results is mainly OK. However, the evaluation metrics is somewhat unclear. It would be better to declare and define the evaluation metrics at the beginning of Section 3.1.
vi) A computational analysis is also required into Section 3.
vii) Could you compare your method to at least one other state-of-the-art method?
viii) In the discussion section, the authors should also mention the possible limitations of the proposed method.
Author Response
Please see the attachment.
My thanks are beyond words for your professional comments!

Reviewer 2 Report
Comments and Suggestions for Authors
In this paper,authors present a cross-video pedestrian tracking algorithm, which introduces the spatial information. The proposed algorithm consists of coordinate features that are used to balance the matching process and linear weighting that is used to improve tracking robustness in cross-video. The method proposed in this paper demonstrates improved success rate in matching target pedestrians and enhances the robustness of continuous pedestrian tracking, surpassing traditional methods in the experimental results.
The whole paper is not bad, however, I still have several concerns about this research and draft:
1. Firstly, I think the process diagrams in the article are too rough, such as Figures 4 and 6, which do not reflect the innovative points and key content of the article. These contents require more detailed illustrations to facilitate readers' understanding.
2. In the discussion part, authors proposed that the clarity and color of the surveillance affect the appearance features of the monitored pedestrians, but by introducing coordinates as pedestrian feature information, the matching process not only follows similarity but also considers the closeness of the pedestrian locations. Please consider providing a deeper comparative analysis with other state-of-the-art methods, highlighting unique aspects of your approach.
3. Exploring additional dimensions of data, such as incorporating real-time environmental variables or pedestrian behavior patterns, could further enhance the novelty of the research.
4. The methodology section is well-detailed, but incorporating more about the algorithm's scalability and performance in varied scenarios (like crowded environments, different lighting conditions, etc.) could be beneficial.
5. For experimental results, it would be valuable to include a broader range of testing scenarios. This could demonstrate the robustness and adaptability of the algorithm.
6. Some learning methods are missing. [1]Deep-IRTarget: An automatic target detector in infrared imagery using dual-domain feature extraction and allocation [2] U2D2Net: Unsupervised Unified Image Dehazing and Denoising Network for Single Hazy Image Enhancement
Comments on the Quality of English Language
NAN
Author Response

(The authors gave the same response as above.)

Round 2
Reviewer 1 Report
Comments and Suggestions for Authors
The manuscript is in much better shape. The authors added a comparison to the state-of-the-art to the manuscript. Further, the presentation was also improved. I think the manuscript can be accepted now.
Author Response
Dear Editor/Reviewer,
Thank you for your prompt and thorough review of our manuscript. We greatly appreciate the time and effort you have dedicated to evaluating our work.
We would like to express our gratitude for your valuable feedback and constructive suggestions throughout the review process. Your insights and recommendations have undoubtedly contributed to the overall improvement of our research. We are grateful for your expertise and guidance, which have significantly enhanced the quality of our work.
Once again, we extend our heartfelt thanks for your positive evaluation and for considering our manuscript for publication.
Sincerely,
Yours.
Reviewer 2 Report
Comments and Suggestions for Authors
Authors answers most of my concerns.
1. The Figure 1 should be modified to clarify the contribution or ideas in the paper. The current version is too simple.
2. From equation 2.1 to 2.3, the method is well known. Just emphasize your porposed method.
NAN
Author Response
Please see the attachment.
We greatly appreciate your valuable feedback provided during the second round of revisions.
